# Structural and molecular rationale for the diversification of resistance mediated by the Antibiotic_NAT family

Peter J. Stogios [1,9], Emily Bordeleau[2,9], Zhiyu Xu[1], Tatiana Skarina[1], Elena Evdokimova[1], Sommer Chou [2], Luke Diorio-Toth [3], Alaric W. D'Souza [3], Sanket Patel[3], Gautam Dantas [3,4,5,6], Gerard D. Wright [2] & Alexei Savchenko [1,7,8 ✉]

The environmental microbiome harbors a vast repertoire of antibiotic resistance genes (ARGs) which can serve as evolutionary predecessors for ARGs found in pathogenic bacteria, or can be directly mobilized to pathogens in the presence of selection pressures. Thus, ARGs from benign environmental bacteria are an important resource for understanding clinically relevant resistance. Here, we conduct a comprehensive functional analysis of the Antibiotic_NAT family of aminoglycoside acetyltransferases. We determined a pan-family antibiogram of 21 Antibiotic_NAT enzymes, including 8 derived from clinical isolates and 13 from environmental metagenomic samples. We find that environment-derived representatives confer high-level, broad-spectrum resistance, including against the atypical aminoglycoside apramycin, and that a metagenome-derived gene likely is ancestral to an *aac(3)* gene found in clinical isolates. Through crystallographic analysis, we rationalize the molecular basis for diversification of substrate specificity across the family. This work provides critical data on the molecular mechanism underpinning resistance to established and emergent aminoglycoside antibiotics and broadens our understanding of ARGs in the environment.

[1] Department of Chemical Engineering and Applied Chemistry, University of Toronto, Toronto M5S 3E5, Canada. [2] David Braley Centre for Antibiotic Discovery, M.G. DeGroote Institute for Infectious Disease Research, Department of Biochemistry and Biomedical Sciences, McMaster University, Hamilton, ON L8S 4L8, Canada. [3] The Edison Family Center for Genome Sciences & Systems Biology, Washington University School of Medicine, St. Louis, MO 63110, USA. [4] Department of Pathology and Immunology, Washington University School of Medicine, St. Louis, MO 63110, USA. [5] Department of Biomedical Engineering, Washington University in St. Louis, St. Louis, MO 63130, USA. [6] Department of Molecular Microbiology, Washington University School of Medicine, St. Louis, MO 63110, USA. [7] Department of Microbiology, Immunology and Infectious Diseases, University of Calgary, Calgary T2N 4N1, Canada. [8] Center for Structural Genomics of Infectious Diseases (CSGID), University of Calgary, Calgary T2N 4N1, Canada. [9] These authors contributed equally: Peter J. Stogios, Emily Bordeleau. ✉email: alexei.savchenko@ucalgary.ca

Antibiotic resistance is a global crisis that threatens every class of clinically deployed antibiotic[1]. Antibiotic resistance genes (ARGs) isolated from bacteria that cause life-threatening disease can often be traced to environmental microbial communities (reviewed in refs. [2–4]). To understand the sources of antibiotic resistance, the identification of links connecting ARGs in the clinic with those in the environment, characterization of their horizontal transfer, evolution, and biochemical/molecular properties are focuses of continuing research (reviewed in refs. [4,5]). For example, the *mcr* family of plasmid-borne colistin resistance genes is thought to originate from chromosomal genes found in various *Moraxella* and *Aeromonas* species[6–8]. The family of extended-spectrum β-lactamase *bla*$_{CTX-M}$ genes found on plasmids of Gram-negative pathogens has been traced to the chromosomal genes of various *Kluyvera* species that are only rarely pathogenic[9]. Given the regular exchange of genetic material harboring ARGs between microbial species, more research is required to understand the breadth and depth of the global resistome, including such aspects as the scope of resistance mechanisms, the specificity and efficiency of ARG products in conferring resistance, and their potential to be mobilized and transferred to pathogens. This comprehensive data is critical for tackling the antibiotic resistance crisis[5,10].

Aminoglycosides (AGs) (Fig. 1) are widely used to treat infections caused both by Gram-positive and Gram-negative bacteria due to their broad-spectrum activity[11]. Toxicity and resistance are

significant problems complicating the use of this class of drugs; nonetheless, they retain value for treating multi-drug and extensively-drug resistant Gram-negative pathogens causing serious infections[12]. Canonical AGs are characterized by a core 2-deoxystreptamine ring with substitutions at the 4- and 6- or 4- and 5- positions. Non-canonical AGs possess variations on the 2-deoxystreptamine core such as streptomycin, or apramycin which contains a fourth ring structure fused to 2-deoxystreptamine. Apramycin is currently used in veterinary medicine[13,14], and with the notable exceptions of *aac3-IV* and the emerging resistance determinant *apmA*[15,16], few ARGs confer resistance to apramycin, prompting excitement for broader deployment in medicine[17–21].

AG resistance is primarily conferred by three classes of aminoglycoside-modifying enzymes (AMEs): phosphotransferases (APHs), nucleotidylyltransferases (ANTs), and acetyltransferases (AACs)[22]. AMEs permanently alter the AG substrate, preventing them from binding to their target, the A-site of the 16S rRNA in the bacterial ribosome. AMEs are widely disseminated in pathogens. Current research focuses on their specificity, mechanisms, and inhibition by small molecules to fortify the design of next-generation AG against resistance, as exemplified by the development of plazomicin and apramycin analogs (apralogs[23–25]).

Previously, we identified 27 AACs in grassland soil microbial communities using a functional metagenomics (FMG) approach[26,27]. These AACs belonged to two sequence and structurally distinct acetyltransferase families—GNAT (GCN5-related

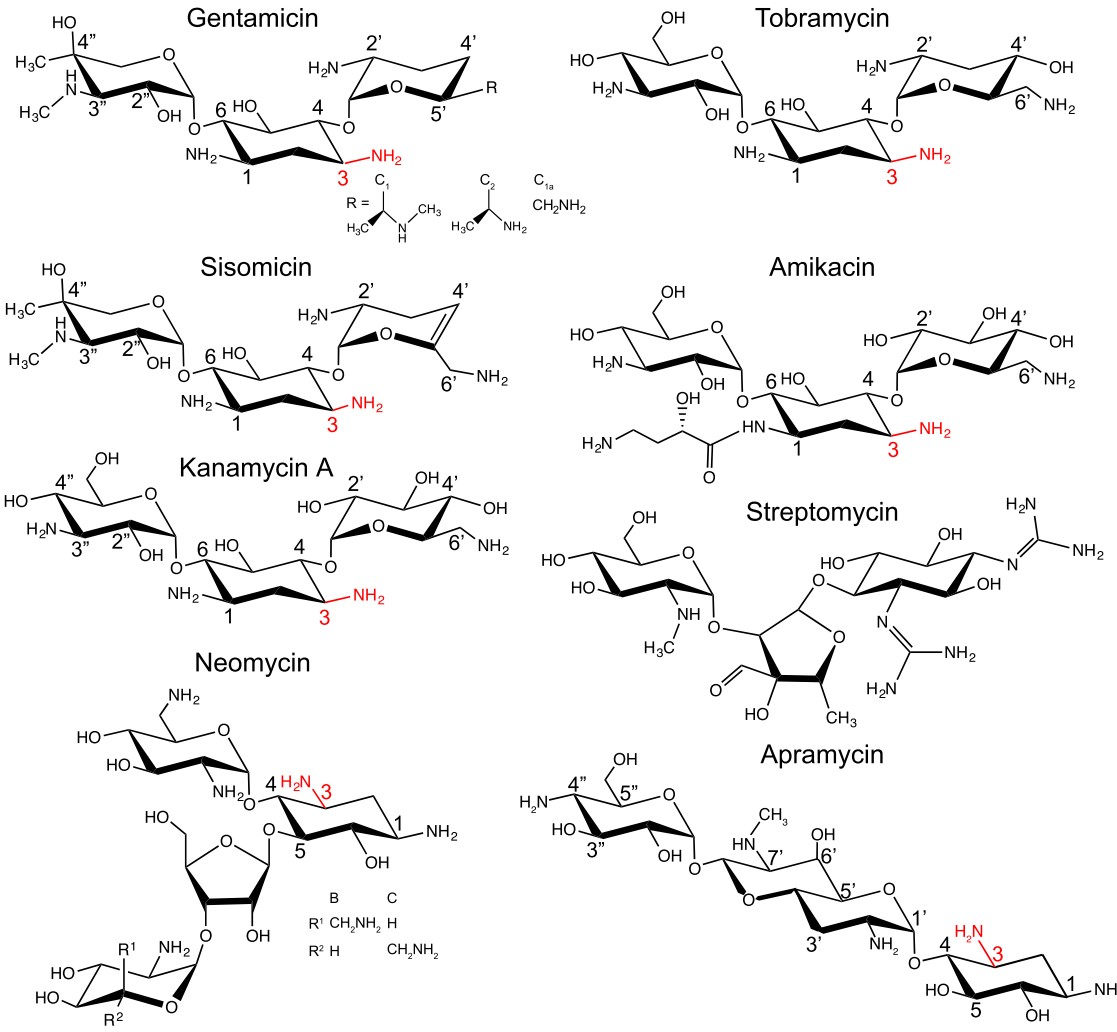

**Fig. 1 Chemical structures of aminoglycosides.** The 3-amino group is highlighted in red.

N-acetyltransferase) and Antibiotic_NAT. These families are distinct in sequence length (approx. 120 residues for GNAT and approx. 220 residues for Antibiotic_NAT) and are classified distinctly by sequence databases (Antibiotic_NAT in Pfam: family Antibiotic_NAT (PF02522), clan Antibiotic_NAT (CL0627) vs GNAT: Acetyltransf_1 (PF00583), Clan Acetyltrans (CL0257)) and by structural databases (Antibiotic_NAT in SCOP: Class = Alpha and beta proteins (a/b), Fold = TTHA0583/YokD-like, Superfamily = TTHA0583/YokD-like, Family Aminoglycoside 3-N-acetyltransferase-like vs GNAT: Class Alpha and beta proteins (a + b), Fold: Acyl-CoA N-acyltransferases (Nat), Superfamily: Acyl-CoA N-acyltransferases (Nat), Family: N-acetyltransferase, NAT). Furthermore, the distinction between these two families is reflected in the divergence in the topology of the β-sheet core of each fold, where the Antibiotic_NAT family is centered on a 3-stranded parallel β-sheet while the GNAT family is centered on a 4-stranded antiparallel β-sheet. Finally, the two families utilize distinct enzymatic mechanisms, with Antibiotic_NAT utilizing a catalytic histidine/glutamate dyad[28] while GNAT utilizes a catalytic tyrosine and glutamate pair[29]. For GNAT AACs, we showed that many environment derived ARGs, which we called meta-AACs for metagenomic AACs, possess resistance activity, acetylation efficiency, and structural properties comparable to AMEs derived from drug-resistant clinical species[24,26]. Our research established that GNAT meta-AACs include all the qualities necessary to cause high-level resistance if mobilized and transferred to human pathogens.

In contrast to the GNAT family, less is known about the biochemical, structural, and molecular features of the Antibiotic_NAT family. There are approximately 50 members of this family identified[26] and many are highly disseminated in Gram-negative pathogens[30], including AAC(3)-II, AAC(3)-III, and AAC(3)-IVa. The AAC(3)-IIa enzyme possesses narrow AG specificity as it is active only against 4,6-disubstituted compounds, while AAC(3)-IIIa is strongly promiscuous due to its activity against a broad range of 4,5- and 4,6-disubstituted AGs[28,31]. The AAC(3)-IVa enzyme was also shown to be promiscuous against a broad range of 4,5- and 4,6-disubstituted AGs as well as against apramycin[15]. There have been no studies describing the enzymatic characteristics of environment-derived members of this family and no comprehensive family-wide analysis to understand their diversification of structure and function.

Several members of the Antibiotic_NAT family have been structurally characterized, including AAC(3)-IIIb and AAC(3)-VIa[28,32] (note: these were erroneously assigned as members of the GNAT family of AAC enzymes in these publications). Other structurally characterized members of this family include FrbF from *Streptomyces rubellomurinus*[33], YokD from *Bacillus subtilis*, and BA2930 from *Bacillus anthracis*[34], none of which possess activity against AGs.

Here, we report a comprehensive structural and functional analysis of the aminoglycoside-resistance spectrum conferred by Antibiotic_NAT family enzymes through characterization of 13 environment-derived enzymes and 8 enzymes derived from clinical isolates. This analysis shows that many confer high-level, broad-spectrum aminoglycoside resistance, and five environment-derived enzymes confer apramycin resistance. Crystallographic analysis of various family members, including meta-AAC0038, AAC(3)-IVa, AAC(3)-IIb, and AAC(3)-Xa, allowed the construction of a molecular model explaining the diversification of substrate specificity in this ARG family.

## Results

**The Antibiotic_NAT family sequences branch into four distinct clades, with all but one including environment-derived members**. Identification of new members of the Antibiotic_NAT family through antibiotic selections of soil metagenomic libraries[35] prompted a revisit of the sequence diversity of this

family. Comparative sequence analysis of the family, including these 14 enzymes derived from environmental microbial communities, 12 Antibiotic_NAT enzymes originating from pathogenic strains, and 25 additional representatives identified by BLAST searches of Genbank, confirmed the presence of conserved sequence motifs typical of Antibiotic_NAT enzymes (Supplementary Fig. 1). This analysis also identified highly variable regions that correspond to residues 62–95, 110–117, 127–142, and 190–212 in meta-AAC0038, along with a variable C-terminal region (Supplementary Fig. 1). The TxΦHΦAE (where Φ = a hydrophobic residue) sequence motif was previously proposed to contain key catalytic residues of this family[28,32,33]. The glutamate residue in this motif interacts with the histidine serving to increase the basicity of the latter residue. The histidine extracts a proton from the AG 3-N-amine group, activating it for nucleophilic attack on the acetyl-CoA carbonyl group[28,32,33]. The threonine in this motif is thought to stabilize the tetrahedral intermediate. Similar sequence signatures (residues Thr165-Glu171 in meta-AAC0038, Supplementary Fig. 1) were identified in all analyzed members of this family, with His and Glu (His168 and Glu171 in meta-AAC0038) along with two glycine residues (Gly122 and Gly158 in metaAAC0038), completely conserved. This motif's threonine (Thr165 in meta-AAC0038) is also conserved in all but one of the analyzed sequences where it is substituted by a chemically similar serine (Supplementary Fig. 1)[28,32,33].

Bayesian reconstruction of the phylogeny of the Antibiotic_NAT family revealed four main clades (Groups 1–4, Fig. 2). Enzymes identified by our metagenomic sampling were distributed among all the clades except for Group 2, which exclusively contains sequences derived from *Actinomycetes*. Several meta-AACs such as meta-AAC0038, meta-AAC0016, and meta-AAC0043 appear to be paralogs of AAC(3)-III, AAC(3)-IVa, and AAC(3)-IIa, respectively.

**Pan-family antimicrobial susceptibility testing aligns substrate specificity with phylogeny**. To comprehensively characterize the spectrum and degree of resistance conferred by Antibiotic_NAT family members, we tested the antimicrobial susceptibility of *Escherichia coli* individually harboring the 21 different genes coding for Antibiotic_NAT enzymes on the pGDP3 plasmid[36]. The results (Fig. 2 and Supplementary Table 1) show that the spectrum and degree of AG resistance correlate with the phylogenetic clustering. Group 1 members including AAC(3)-IVa and four meta-AACs confer the broadest spectrum and highest degree of resistance to 4,6- and 4,5-disubstituted AGs, consistent with previous studies on AAC(3)-IVa[15], and confer high-level resistance to apramycin. We found that the Group 2 member AAC(3)-Xa, derived from an Actinomycetes, is limited in its AG specificity to the 4,6-disubstituted AGs kanamycin and tobramycin; the only other Group 2 member tested in our host *E. coli* was AAC(3)-IXa and did not convey any detectable AG resistance. Group 3 enzymes including AAC(3)-IIIb and four meta-AACs confer resistance to 4,6- and 4,5-disubstituted AGs, consistent with previous data reported for AAC(3)-III enzymes[28,31]; meta-AAC0038 is the lone member of this family that confers resistance to apramycin. Group 4 members are restricted in activity to 4,6-disubstituted AGs, including AAC(3)-IIb/IIc and six meta-AAC enzymes, which is reflective of reports on the resistance profile of AAC(3)-VIa[32,37]; AAC(3)-IIb also confers low-level apramycin resistance.

Notably, each meta-AAC confers AG resistance, with many demonstrating broad-spectrum and high-level resistance, including against apramycin (meta-AAC0016, meta-AAC0018, meta-AAC0033, meta-AAC0030, and meta-AAC0038).

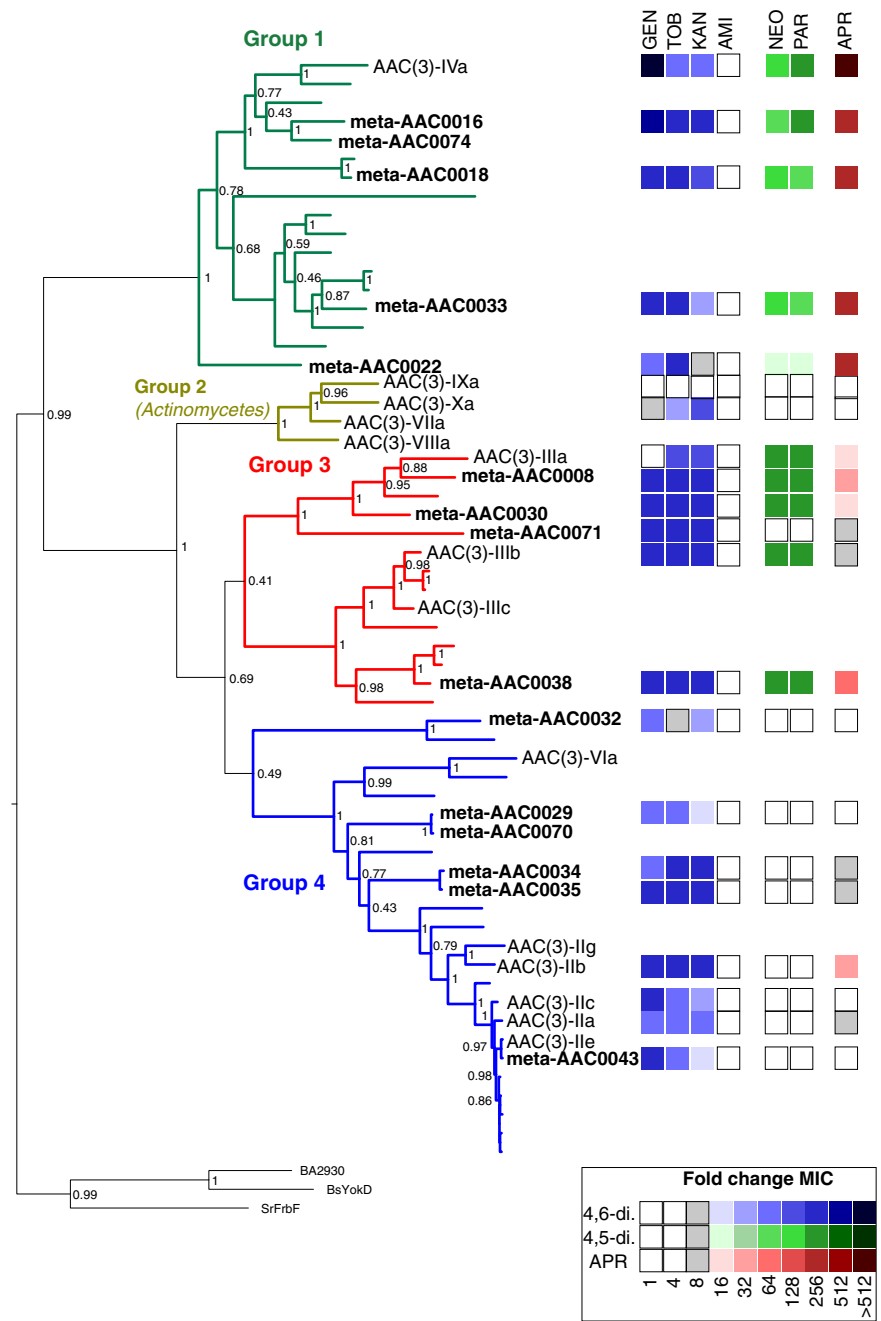

**Fig. 2 Family-wide antibiotic susceptibility mapped onto phylogenetic reconstruction of Antibiotic_NAT family.** The four main groups are separately colored. Sequence names are only shown for meta-AAC, clinical isolates of AAC(3) enzymes, and outgroup members with Antibiotic_NAT fold but with no activity against aminoglycosides (FrbF, YokD, and BA2930); other sequences not labeled are hits from a BLAST search of NCBI nr database. Node labels are Bayesian probability values. The right side represents a heatmap of AG susceptibility (fold change MIC relative to control strain containing no resistance element), with key shown at bottom right, full data in Supplementary Table 1.

**Crystal structures of meta-AAC0038, AAC(3)-IVa, AAC(3)-IIb, and AAC(3)-Xa enzymes show that the variation in the minor subdomain is responsible for diversity in activity against AGs.** We undertook a structural genomics campaign to understand the structural basis of the evident diversification of substrate specificity across the Antibiotic_NAT family, with a particular interest in the broadly active Group 1 and meta-AAC enzymes. We solved crystal structures of the AAC(3)-IVa, AAC(3)-IIb, AAC(3)-Xa, and meta-AAC0038 enzymes, including ligand-bound states of AAC(3)-IVa and meta-AAC0038. Crystallographic statistics for all determined structures are shown in Table 1.

The fold typical of the Antibiotic_NAT family is evident in all structures, composed of 13 α-helices and 8 β-strands (Fig. 3a), and determined structures superpose with pairwise RMSD's 0.8–1.0 Å between 197 to 266 matching Cα atoms. Notably, the primary sequence most conserved across the family representatives (Supplementary Fig. 1) belongs to what we defined as a major subdomain in the Antibiotic_NAT fold (Fig. 3b). In contrast, the variable sequence regions identified by our comparative analysis (see above) constitute a minor subdomain (Fig. 3b). According to this distinction, the major subdomain is centered on a 7-stranded antiparallel β-sheet with a bundle of 5 α-helices arranged on one face of the sheet, with the second bundle

**Table 1 X-ray crystallographic statistics.**

| Structure | Meta-AAC0038[H29A] apoenzyme | Meta-AAC0038[H168A]•AcCoA | Meta-AAC0038[H168A]•CoA | Meta-AAC0038[H168A]•apramycin•CoA | |
|---|---|---|---|---|---|
| *PDB code* | 6MMZ | 6MN0 | 5HT0 | 7KES | |
| *Data collection* | | | | | |
| Space group | C2 | C2 | C2 | P3$_1$2$_1$ | |
| Cell dimensions | | | | | |
| $a$, $b$, $c$ (Å) | 105.8, 158.1, 143.4 | 108.1, 159.6, 143.3 | 107.02, 159.50, 146.22 | 127.77, 127.77, 94.65 | |
| $\alpha$, $\beta$, $\gamma$, (°) | 90, 94.9, 90 | 90, 94.6, 90 | 90, 94.7, 90 | 90, 90, 120 | |
| Resolution, Å | 25.00-3.30 | 25.00-2.40 | 25.0-2.75 | 30.0-2.36 | |
| $R_{merge}$[a] | 0.268 (0.743)[b] | 0.094 (0.372) | 0.074 (0.440) | 0.091 (1.427) | |
| $R_{pim}$[c] | 0.142 (0.395) | 0.062 (0.249) | 0.085 (0.251) | 0.031 (0.505) | |
| CC$_{1/2}$ | 0.809[b] | 0.949 | 0.968 | 0.601 | |
| $I / \sigma(I)$ | 6.3 (2.3) | 10.75 (2.09) | 17.76 (3.19) | 21.87 (1.0) | |
| Completeness, % | 99.4 (99.9) | 99.9 (100) | 96.7 (90.4) | 100 (100) | |
| Redundancy | 4.6 (4.6) | 3.3 (3.2) | 4.0 (3.7) | 9.9 (8.8) | |
| *Refinement* | | | | | |
| Resolution, Å | 19.75-3.30 | 24.93-2.39 | 24.97-2.75 | 29.19-2.36 | |
| No. of unique reflections: working, test | 35,122, 1646 | 94,715, 1996 | 60,061, 2021 | 36,879, 1846 | |
| $R_{work}/R_{free}$[d] | 20.4/26.1 (29.6/38.9) | 17.8/20.8 (23.1/28.9) | 20.4/23.3 (31.3/30.8) | 19.1/22.8 (29.6/34.7) | |
| No. of atoms and molecules | | | | | |
| Protein | 11,977; 6 | 12,033; 6 | 12,016; 6 | 3992; 2 | |
| Aminoglycoside | N/A | N/A | N/A | 73, 2 | |
| Acetyl-CoA/CoA | N/A | 306, 6 | 288, 6 | 96, 2 | |
| Solvent | 104 | 236 | 105 | 25 | |
| Water | 104 | 1706 | 343 | 170 | |
| *B*-factors | | | | | |
| Protein | 59.2 | 32.9 | 54.1 | 70.9 | |
| Aminoglycoside | N/A | N/A | N/A | 129.0 | |
| Acetyl-CoA/CoA | N/A | 33.1 | 52.9 | 61.9 | |
| Solvent | 96.2 | 71.4 | 108.2 | 100.5 | |
| Water | 20.4 | 43.4 | 47.7 | 64.2 | |
| R.m.s. deviations | | | | | |
| Bond lengths, Å | 0.002 | 0.005 | 0.014 | 0.005 | |
| Bond angles, ° | 0.552 | 1.770 | 1.827 | 1.337 | |
| **Structure** | **AAC(3)-IVa apoenzyme** | **AAC(3)-IVa[H154A]•APR** | **AAC(3)-IVa[H154A]•GEN** | **AAC(3)-IIb** | **AAC(3)-Xa** |
| *PDB code* | 6MN3 | 6MN4 | 6MN5 | 7LAO | 7LAP |
| *Data collection* | | | | | |
| Space group | C2 | P2$_1$2$_1$2$_1$ | P2$_1$2$_1$2$_1$ | P2$_1$2$_1$2$_1$ | P6$_3$22 |
| Unit cell | | | | | |
| $a$, $b$, $c$ (Å) | 114.2, 55.3, 94.3 | 77.6, 103.5, 264.9 | 77.6, 131.9, 266.9 | 43.2, 61.4, 112.0 | 161.5, 161.5, 138.7 |
| $\alpha$, $\beta$, $\gamma$, (°) | 90, 102.6, 90 | 90, 90, 90 | 90, 90, 90 | 90, 90, 90 | 90, 90, 120 |
| Resolution, Å | 30.00-2.39 | 30.00-2.80 | 40.0-2.58 | 40.0-1.92 | 50.0-2.04 |
| $R_{merge}$ | 0.141 (0.997) | 0.183 (1.771) | 0.086 (0.542) | 0.080 (0.332) | 0.098 (1.074) |
| $R_{pim}$ | 0.080 (0.572) | 0.063 (0.613) | 0.048 (0.371) | 0.037 (0.159) | 0.024 (0.396) |
| CC$_{1/2}$ | 0.524 | 0.776 | 0.703 | 0.524 | 0.593 |
| $I / \sigma(I)$ | 9.98 (1.25) | 12.77 (1.40) | 14.08 (1.08) | 26.19 (3.13) | 31.42 (1.08) |
| Completeness, % | 98.8 (99.9) | 95.2 (97.1) | 95.7 (82.0) | 95.8 (79.8) | 99.5 (92.9) |
| Redundancy | 3.9 (3.9) | 9.0 (8.9) | 3.7 (2.5) | 5.4 (4.7) | 17.2 (6.6) |
| *Refinement* | | | | | |
| Resolution, Å | 30-2.39 | 29.33-2.80 | 38.4-2.58 | 35.32-1.92 | 49.39-2.04 |
| No. of unique reflections: working, test | 22,587, 1129 | 63,643, 3627 | 83,240, 2000 | 22,528, 2167 | 66,736, 3279 |
| $R$-factor/free $R$-factor | 18.2/22.8 (27.1/32.2) | 26.1/32.1 (35.3/41.1) | 18.7/22.1 (26.8/28.1) | 18.0/22.9 (23.6/29.4) | 16.5/19.8 (28.7/31.5) |
| No. of refined atoms and molecules | | | | | |
| Protein | 3921; 2 | 11,564; 6 | 11,824; 6 | 2045; 1 | 4421; 2 |
| Aminoglycoside | N/A | 186, 5 | 186, 6 | N/A | N/A |
| Acetyl-CoA/CoA | N/A | N/A | N/A | N/A | N/A |
| Solvent | 3 | 5 | 422 | 17 | 58 |
| Water | 176 | 330 | 519 | 211 | 678 |

**Table 1 (continued)**

| Structure | Meta-AAC0038$^{H29A}$ apoenzyme | Meta-AAC0038$^{H168A}$•AcCoA | Meta-AAC0038$^{H168A}$•CoA | Meta-AAC0038$^{H168A}$•apramycin•CoA | |
|---|---|---|---|---|---|
| B-factors | | | | | |
| Protein | 48.7 | 90.4 | 91.2 | 50.8 | 53.1 |
| Aminoglycoside | N/A | 111.0 | 142.9 | N/A | N/A |
| Acetyl-CoA/CoA | N/A | N/A | N/A | N/A | N/A |
| Solvent | 53.3 | 86.7 | 98.0 | 70.3 | 105.8 |
| Water | 39.6 | 61.4 | 72.6 | 45.7 | 63.2 |
| R.m.s.d. | | | | | |
| Bond lengths, Å | 0.004 | 0.006 | 0.006 | 0.014 | 0.012 |
| Bond angles, ° | 0.803 | 1.071 | 0.879 | 1.260 | 1.118 |

ND not determined.

[a]$R_{merge} = \Sigma_{hkl}\Sigma_j |I_{hkl,j} - \langle I_{hkl}\rangle|/\Sigma_{hkl}\Sigma_j I_{hkl,j}$, where $I_{hkl,j}$ and $\langle I_{hkl}\rangle$ are the jth and mean measurement of the intensity of reflection j.

[b]All values in brackets and all $CC_{1/2}$ values refer to the highest resolution shells.

[c]$R_{pim} = \Sigma_{hkl}\sqrt{(n/n-1)}\ \Sigma_{j=1}^{\eta}|I_{hkl,j} - \langle I_{hkl}\rangle|/\Sigma_{hkl}\Sigma_j I_{hkl,j}$.

[d]$R = \Sigma|F_p^{obs} - F_p^{calc}|/\Sigma F_p^{obs}$, where $F_p^{obs}$ and $F_p^{calc}$ are the observed and calculated structure factor amplitudes, respectively.

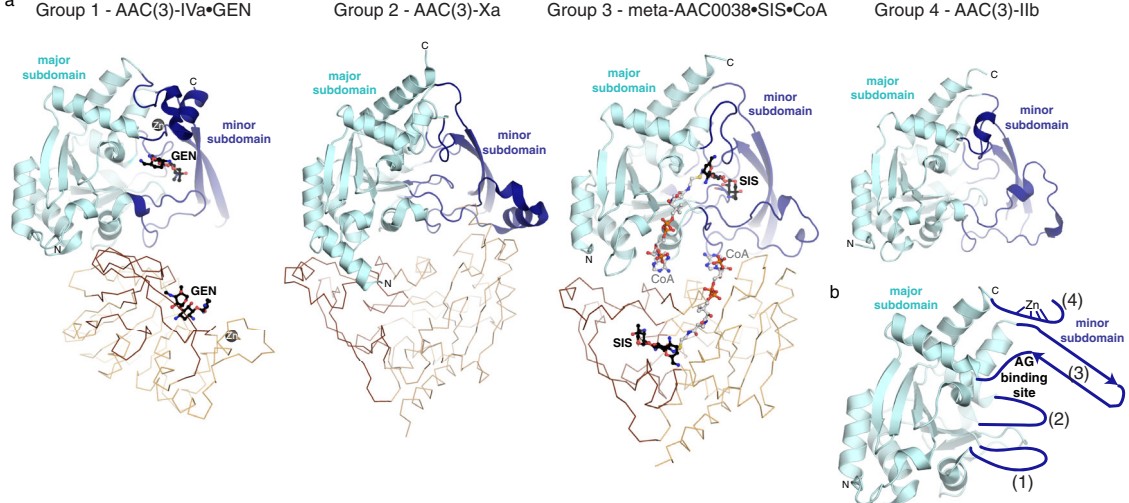

**Fig. 3 Structural analysis of Antibiotic_NAT enzymes. a** Structures of AAC(3)-IVa, AAC(3)-Xa, meta-AAC0038, and AAC(3)-IIb as representatives of groups 1–4, respectively. The conserved major subdomain of the Antibiotic_NAT fold is colored in cyan; the variable minor subdomain is colored in dark blue. The second subunit in the AAC(3)-IVa, AAC(3)-Xa and meta-AAC0038 crystal structures are shown in thin orange lines. $Zn^{2+}$ ion bound to AAC(3)-IVa is shown as a dark gray sphere. Ligands bound to AAC(3)-IVa and meta-AAC0038 are shown in sticks and labeled. **b** Schematic of structural variations in the minor subdomain as insertions or extensions to the major subdomain, numbered 1–4.

of 4 α-helices arranged on the other face of the sheet. The minor subdomain is characterized by four main structural variations that are subfamily-specific, which we called inserts 1–4. Insert 1 (Fig. 3b) forms an extended loop structure of variable length while adopting a helical structure in AAC(3)-Xa, meta-AAC0038, and AAC(3)-IIb but not in AAC(3)-IVa. Insert 2 forms a short turn between two α-helices, which most closely impacts the AG binding site. Insert 3 forms a two-stranded antiparallel β-sheet while corresponding to a short α-helix found only in AAC(3)-IIb structure. Finally, insert 4 is a C-terminal extension to the major subdomain unique to AAC(3)-IVa and forms an α-helix and a C3H1 $Zn^{2+}$ binding site. Altogether, this global structural analysis reflects that the minor domain is the principal source of structural diversity among members of this family. A negatively charged cleft is formed in the region between the minor and major subdomains in each structure, with the deepest section formed primarily by the minor subdomain. As will be discussed in detail later, this cleft harbors the AG binding site.

The Antibiotic_NAT enzymes also diversify in their oligomerization state. The meta-AAC0038 adopts a dimeric structure with a buried surface of ~900 Å² per subunit (Fig. 3). This enzyme also

forms a dimer in solution according to the size exclusion chromatography (not shown). In contrast, the AAC(3)-Xa enzyme exists as a monomer in solution despite forming a dimer in the crystal lattice (Fig. 3). AAC(3)-IVa also adopted a dimeric structure (Fig. 3) both in crystal and in solution, in line with previous reports on its oligomeric state[15], but the arrangement of the two chains in this enzyme differed from that of the meta-AAC0038 dimer. The buried surface area between subunits of the AAC(3)-IVa dimer (~650 Å²) was formed nearly exclusively through interactions between the major subdomains of the two monomers of this enzyme. Finally, AAC(3)-IIb was monomeric both in the crystal structure and in solution (not shown).

**Structural analysis of the group 1 enzyme AAC(3)-IVa suggests a mechanism for broad specificity against AG substrates.** To understand the structural basis of the highly promiscuous nature of group 1 Antibiotic_NAT enzymes, we pursued structural characterization of the AAC(3)-IVa representative of this clade in complex with AG substrates. To increase the chances of capturing substrate-bound enzyme complex we used the catalytically impaired His154Ala mutant of AAC(3)-IVa.

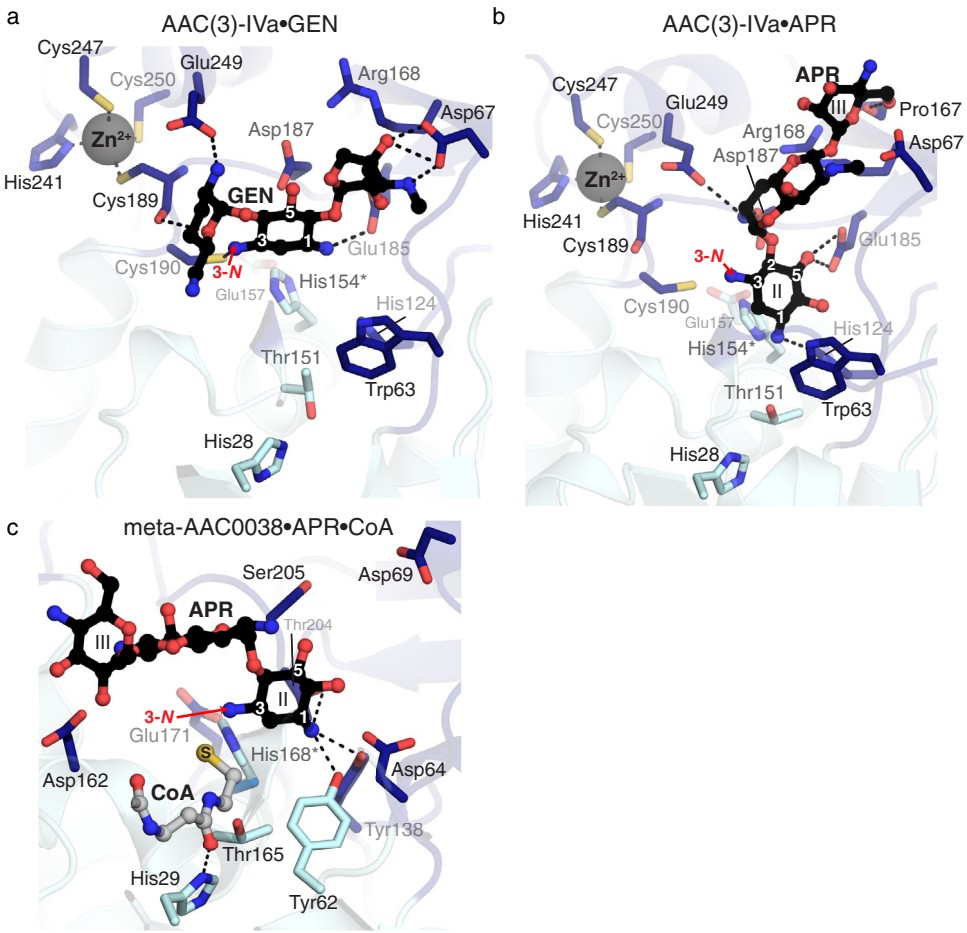

**Fig. 4 Details of molecular recognition of aminoglycosides by meta-AAC0038 and AAC(3)-IVa. a** From solved crystal structures, active sites of AAC(3)-IVa[H154A] and gentamicin, **b** AAC(3)-IVa[H154A] and apramycin, and **c** meta-AAC0038[H168A] and apramycin and CoA. Dashes indicate hydrogen bonds. Since each protein was crystallized with inactive mutants, His168Ala of His154A mutations for meta-AAC0038 and AAC(3)-IVa, respectively, these sidechains shown in this figure are from the apoenzyme structures and indicated with asterisks. Residues colored in dark and light blue are from the major and minor subdomains of the two enzymes, respectively. Acetylation sites (3-N groups) are labeled with red arrows.

Using this strategy, we were able to determine the crystal structures of AAC(3)-IVa enzyme in complex with gentamicin or apramycin to 2.6 and 2.8 Å, respectively. In both complex structures, the electron density corresponding to the AG molecule localized to the cleft between the major and minor subdomains of the enzyme. Most of the AG substrate interactions with the protein are mediated by amino acid sidechains from the minor subdomain (Fig. 4a–c). For the AG substrate in both structures, the 3-N group is positioned close to residue 154 and proximal to the presumed location of the thiol of CoA. We observe a similar substrate orientation in the crystal structures of meta-AAC0038 enzyme complexes, described below, suggesting a common active site topology for this family.

In the complex structures, the gentamicin molecule spans across the enzyme's minor subdomain while the apramycin molecule is twisted nearly 90° relative to gentamicin. This difference is reflected in the rotation of the 2-deoxystreptamine rings of each compound (Fig. 4b). The 2-deoxystreptamine/II ring of apramycin stacks against the sidechain of Trp63, and its rotation positioned the central and III rings more into the minor subdomain cleft and towards Asp67. Notably, these two residues are contributed from the much shorter hairpin connecting the α4 and α5 helices compared to the equivalent region in the other enzymes we crystallized. Additionally, Glu185 appears to be a critical residue for interactions with gentamicin and apramycin as

it positions the 2-deoxystreptamine ring for modification through interactions with the 1-N of gentamicin or the 5-hydroxyl of apramycin. Interestingly, Cys190, which is just N-terminal to the Zn²⁺ binding site, interacts with the 3-N of gentamicin. Finally, the C-terminal extension of AAC(3)-IVa corresponding to residues 236-257 contributes to the interactions with both gentamicin and apramycin via Glu249 side chain.

We identified a Zn²⁺ ion binding site in the C-terminal extension of AAC(3)-IVa structure. This feature may be of only structural significance since neither this ion nor the sidechains of its cysteine and histidine ligands formed any interactions with the AGs. The binding of Zn²⁺ could stabilize this region and allow for orientation of the Glu249 residue for AG recognition. The Zn²⁺-binding residues are fully conserved across Antibiotic_NAT Group 1 representatives.

The analysis of the AAC(3)-IVa•gentamicin complex allowed us to propose a mechanism for this enzyme's ability to recognize 4,5-disubstituted AGs. In the complex structure, gentamicin's 5-OH pointed out of the enzyme's active site. If similarly oriented, 4,5-disubstituted AGs would not cause a steric clash with this enzyme's active site. Collectively, these observations show that AGs can adopt multiple bound orientations facilitated by the dramatic structural changes in the minor subdomain of AAC(3)-IVa, thereby supporting broad substrate specificity for AG modification.

**The meta-AAC0038 enzyme active site's molecular architecture allows for activity against 4,5 and 4,6-disubstituted AGs.** Our data presented above demonstrated that the environmental metagenome-derived meta-AAC0038 enzyme can confer high and broad resistance to AGs including to the atypical AG apramycin when expressed in *E. coli*. Using the catalytically inactive His168Ala mutant of this enzyme, we were able to determine the crystal structures of ternary meta-AAC0038$^{H168A}$•apramycin•CoA and the binary meta-AAC0038$^{H168A}$•acetyl-CoA complexes.

In line with the previously discussed Antibiotic_NAT enzyme structures, meta-AAC0038 accommodated the substrates in the negatively charged cleft formed by the minor subdomain, with the 3-*N* group of apramycin located within 2.6 Å of the sulfhydryl group of CoA (Fig. 4c). Notably, the I and III rings of apramycin were positioned out from the active site cleft and did not form interactions with the enzyme except for hydrogen bonds with the Asp94 and Asp162 sidechains. The ability to retain this AG molecule in the active site via very few contacts could explain the activity of meta-AAC0038 on this substrate resulting in the low-level resistance to apramycin which was not detected for the other representatives of Group 3 Antibiotic_NAT enzymes.

AAC(3)-IIIb, another group 3 enzyme, has been previously characterized in detail for its interactions with 4,6- and 4,5-disubstituted AGs[28]. The meta-AAC0038 and AAC(3)-IIIb structures superimpose with RMSD 0.54 Å across 219 Cα atoms, share all the minor subdomain structural elements, and show complete conservation of AG binding residues (Fig. 4c). However, the position corresponding to Glu223 in AAC(3)-IIIb is occupied by Asp213 in meta-AAC0038. Glu223 is positioned at the ring I binding site of apramycin, which may impact the ability of AAC(3)-IIIb to accommodate this AG as a substrate.

**The group 4 enzyme AAC(3)-IIb harbors a restricted active site.** The crystal structure of AAC(3)-IIb represents the first molecular image of enzymes with AAC(3)-II activity. Its structure superimposes with RMSD 0.7 Å over 221 Cα atoms with the previously characterized AAC(3)-VIa structure[32], consistent with our phylogenetic analysis placing both these enzymes in the group 4 of the Antibiotic_NAT family. Similarly to the AAC(3)-VIa enzyme[32], the minor subdomain loop of AAC(3)-IIb contains the conserved Asn208, which is predicted to clash with substituents at position 5 of the AG substrate, thereby explaining the lack of activity toward 4,5-disubstituted AGs. Other notable amino acids in the active site of AAC(3)-IIa that may restrict the size and positioning of AG substrates include Tyr66, positioned near the binding location of the double prime ring (Fig. 1), and Phe97, positioned near the central 2-deoxystreptamine ring. Altogether, AAC(3)-IIb—like AAC(3)-VIa—harbors a more restricted active site, consistent with its limited AG specificity.

**AAC(3)-Xa also harbors a restricted AG binding site.** As indicated by our AG susceptibility testing, the activity of AAC(3)-Xa is limited to tobramycin and kanamycin (Fig. 2). To rationalize this strict specificity, we modeled the position of kanamycin into the active site of the apoenzyme structure based on the position of gentamicin bound to AAC(3)-IVa. This analysis suggested that gentamicin would not be accommodated due to the Tyr79 and Asp130 residues, which would clash with the 4"-OH group or the methylated 3"-amine of the corresponding AG substrate, respectively. This model also provides a hypothesis for the inability of this enzyme to confer resistance to 4,5-disubstituted AGs, as the 5-substituents would clash with Glu220 of the enzyme. Based on comparative analysis of the AAC(3)-Xa and AAC(3)-IVa•apramycin complex structures, Tyr79 would also introduce a steric clash with this AG in

the AAC(3)-Xa active site. Notably, Tyr79, Asp130, Glu220, and adjacent active site residues are highly conserved in Antibiotic_NAT Group 2 (Supplementary Fig. 1), suggesting these are critical determinants for restricting the specificity of these enzymes.

**Genetic elements adjacent to meta-AACs suggest possible mobilization mechanisms.** To investigate the potential for lateral transfer of meta-AACs, we searched for mobile genetic elements (MGEs) on the AAC-encoding contigs. Of the genes recovered through FMG, only one - meta-AAC0043 - is syntenic with multiple MGEs. This sequence is co-localized on our phylogeny (Fig. 1) with *aac(3)-IIe*, suggesting a close evolutionary relationship. This finding is in line with the observation that all 28 gentamicin-selected FMG contigs annotated with a gene encoding an AAC(3)-II family enzyme were syntenic with at least one MGE. Worryingly, this contig shows extremely high similarity to sequences found in both chromosomes and plasmids of pathogens like *E. coli*, *K. pneumoniae*, *C. freundii*, and *V. cholerae* (Fig. 5). Taken together, our analysis demonstrates that representatives of Antibiotic_NAT family encoded by the environmental microbiome can be directly mobilized across taxonomic boundaries to convey resistance in clinically important bacterial species.

## Discussion

The realization that environmental microbial communities are important reservoirs of ARGs provides keys to understanding the emergence of antibiotic resistance in pathogenic species. For most ARG families, the evolution, transferability, and molecular/structural basis for the activity of their environmental relatives has not been well characterized. Given that antibiotic use in agricultural and other anthropogenic settings represents a significant proportion of global antibiotic deployment, it is vital to understand the scope and breadth of resistance in the broader global resistome, which may select for the evolution and transfer of ARGs. This knowledge is critical to protecting the potency of our current antibiotic arsenal and designing antibiotics that are less susceptible to ARGs.

In this study, we follow on our previous identification of multiple Antibiotic_NAT family members in soil-derived metagenomic libraries[35] through detailed structural and functional analysis. Firstly, the phylogenetic reconstruction of this family that we calculated was linked to a comprehensive study of the substrate specificity profiles of the four main clades, represented by the AAC(3)-IV, AAC(3)-VII/VIII/IX/X, AAC(3)-III, and AAC(3)-II/IV enzymes. Secondly, with the additional crystal structures described in this study and comparison to previously-available structural information, we conclusively show that this division is reflected in differences in activity against AG substrates and in structural diversification localized to the minor subdomain of the Antibiotic_NAT fold. Given that the minor subdomain is much less conserved between Antibiotic_NAT family members, the deficit in molecular information about variations in this subdomain that would allow for a better understanding of the role of individual amino acids in this region for substrate specificity necessitated and inspired our structural investigation into additional representatives of this family. Thirdly, we show that environment-derived enzymes of this family, which previously have not been characterized for molecular determinants behind their activity against antibiotic substrates, possess resistance-conferring activities comparable to and sometimes exceeding those activities of their counterparts derived from clinical isolates. Fourthly, we show that numerous members of this family inactivate apramycin, an atypical AG that is increasingly being considered for clinical deployment and for which little has been known about possible resistance determinants.

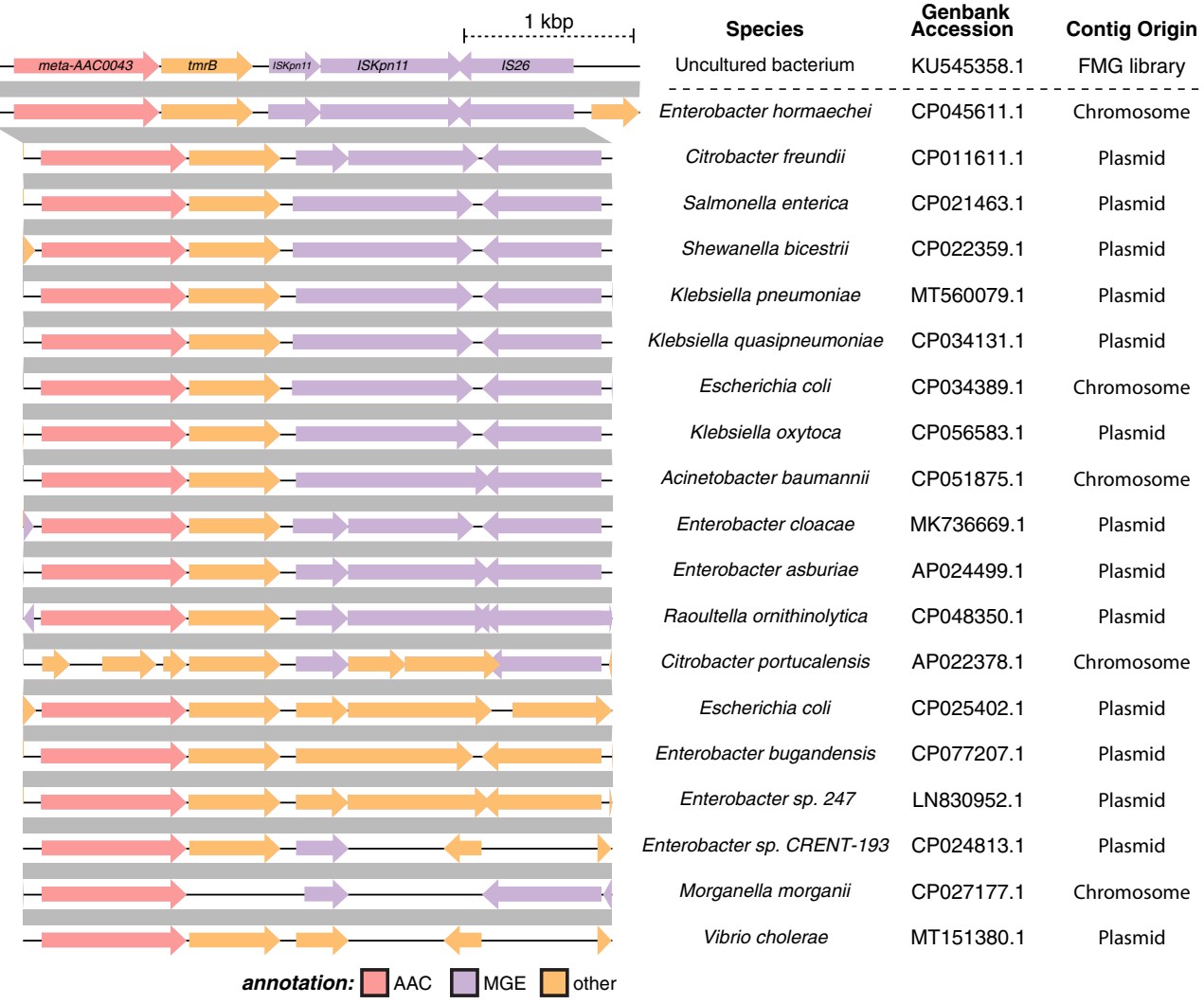

**Fig. 5 Synteny of meta-AAC0043 with mobile genetic elements.** The contig containing meta-AAC0043 was queried against the NCBI nucleotide database and filtered for highly similar sequences, revealing the presence of similar sequences in a hugely diverse set of taxa. A representative set of similar genomic segments are shown, with gray bars indicating blastn percent identity ≥99.5%. Many of these matches are from plasmid sequences, and almost all of them contain ORFs annotated as MGEs (e.g., transposons, insertion sequences, etc.).

Our structural data includes the crystal structure of the AAC(3)-IVa enzyme which is the first molecular image of a Group 1 Antibiotic_NAT enzyme. Our extensive structural and functional characterization demonstrates that this enzyme mediates broad-spectrum AG resistance, including to 4,5-, 4,6-disubstituted AGs and the atypical AG apramycin by evolving a more spacious active site. This is achieved by a C-terminal extension and modifications of the structure and residue composition of the α4-α5 hairpin of the minor subdomain of the enzyme which allows for broad spectrum of AG recognition. The role of the $Zn^{2+}$-binding site in the mechanism of action of AAC(3)-IVa and Group 1 enzymes is the subject of ongoing investigation. After the structures of AAC(3)-IVa•gentamicin and AAC(3)-IVa•apramycin were publicly available in the PDB, another group performed structure-guided mutagenesis on the enzyme[38]. This analysis confirmed the Glu185 and Asp187 residues' important roles for interactions with AG substrates, and the role of the Asp67 residue in specificity for gentamicin recognition. This group also generated a double mutant Cys247Ser/Cys250Ser, which abrogated resistance to both gentamicin and apramycin, suggesting that $Zn^{2+}$-binding is necessary for substrate recognition. However, since no evidence for the effect of these two mutations on the overall stability of this enzyme was

provided, the direct effect of $Zn^{2+}$ binding on interaction with AG substrates remains unclear.

According to our sequence analysis the Group 1 members meta-AAC0022, meta-AAC0033, meta-AAC0016, and meta-AAC0018 also share the C-terminal extension, the $Zn^{2+}$-binding residues, and the shorter sequence corresponding to the α4-α5 hairpin. We showed that these enzymes are also active against the wide range of AGs including apramycin.

Antibiotic_NAT Group 3 members showed a high degree of promiscuity, including activity toward the 4,5- and 4,6-disubstituted AGs. Notably, the meta-AAC0038 enzyme was also active against apramycin which inspired our structural analysis of this activity. According to our meta-AAC0038-apramycin complex structure, the binding of apramycin to this enzyme differed from its interactions to AAC(3)-IVa. Meta-AAC0038 demonstrated activity analogous to AAC(3)-IIIb and AAC(3)-IIIc enzymes, which belonged to the same clade. Other environment-derived members, including meta-AAC0008, meta-AAC0030, and meta-AAC0071, were similarly active against 4,5- and 4,6-disubstituted AGs.

Representatives of Antibiotic_NAT Groups 2 and 4 were the most restricted in their specificity, and this was reflected in more

constrained and smaller active sites, as revealed by the structures of AAC(3)-IIb and AAC(3)-Xa. The environment-derived enzymes of Group 4, including meta-AAC0032, meta-AAC0029, meta-AAC0034, meta-AAC0035, and meta-AAC0043, likewise conferred resistance only to kanamycin and tobramycin. The crystal structure of AAC(3)-IIb features an active site highly like that of AAC(3)-VIa, consistent with the 4,6-disubstituted specificity of Group 4 enzymes.

Additionally, our study expanded the repertoire of AMEs active against apramycin to include six environment-derived enzymes, with the Group 1 members meta-AAC0016, meta-AAC0018, meta-AAC0033, and meta-AAC0022 conferring high-level apramycin resistance. The presence of these enzymes in environmental microbial species may be provoked by widespread apramycin use in agriculture settings. As apramycin is deployed in the clinic, it is important to be mindful of the possible further dissemination of these ARGs.

Our analysis of lateral gene transfer signatures in the genetic vicinity of meta-AAC genes indicates that these genes show low potential for mobilization, for the most part, with the notable exception of *meta-AAC0043*. This conclusion is corroborated by the separation of meta-AAC and AAC(3) enzyme sequences in each group within our phylogenetic reconstruction, except for the close clustering of meta-AAC0008 with AAC(3)-IIIa (67% identical at the protein level) and meta-AAC0043 with AAC(3)-IIe (96% identical). While no MGEs were identified in the contig containing the *meta-AAC0008* gene, multiple MGEs were present in the contig harboring *meta-AAC0043*. This proximity strongly suggests that *meta-AAC0043* has mobilized into pathogens, manifesting in the enzyme AAC(3)-IIe, conferring resistance to 4,6-disubstituted AGs. This precedent suggests that with further FMG sampling, additional meta-AAC genes may be identified which represent environmental sources of clinically relevant Antibiotic_NAT genes.

The metagenomic, structural, and functional data presented in this study establishes key molecular insights into the molecular basis for AG recognition by all four clades of the Antibiotic_NAT family. This provides a deeper understanding of the primary sequence signatures important for the AG resistance profile conferred by the corresponding enzymes. Our observation that environmental members of this family can confer broad, high-level AG resistance and have already mobilized into pathogenic species warrants surveillance and FMG sampling to detect new connections between ARGs in the clinic and the environment.

## Methods

**Sequence analysis and phylogenetic reconstruction.** Previously identified members of the Antibiotic_NAT family from functional selections of soil metagenomes[35] were aligned with clinically isolated AAC(3) enzyme sequences and homologs in Genbank identified by BLAST. Sequence alignment was performed using the Clustal Omega server (EMBL-EBI). The phylogenetic reconstruction was generated from the sequence alignment by MrBayes[39] (with gamma-distributed rates across sites, rate matrix = mixed, 1,000,000 generations for mcmc) and visualized by using FigTree v1.4.2.

**Antibiotic susceptibility testing.** Environmental and clinical Antibiotic_NAT sequences were cloned into the low copy plasmid pGDP3. Expression levels of each gene were controlled by the strong, constitutive promoter $P_{bla}$. Aminoglycoside susceptibility testing was completed in technical triplicate, single colony dilution replicated across three rows of the same microtiter plate, with our hyperpermeable, efflux-deficient strain *E. coli* BW25113 $\Delta tolC \Delta bamB$ following the Clinical and Laboratory Standards Institute (CLSI) protocols for the microbroth dilution method[40]. *E. coli* was cultured in a cation-adjusted Mueller Hinton broth (CAMHB) arrayed in a 96-well format. The plates were incubated for 18 h at 37 °C. A Labcyte Echo 550 and Thermo Combi nL was used for dispensing the antibiotics and a Formulatrix Tempest for culture dispensing.

**Protein purification.** *E. coli* BL21(DE3) Gold was used for *meta-AAC0038* and *aac(3)-IVa* overexpression. 3 mL overnight culture was diluted into 1 L LB media containing selection antibiotic ampicillin and grown at 37 °C with shaking. The cell

culture was induced with IPTG at 17 °C once the $OD_{600}$ reached 0.6-0.8. Cell pellets were collected by centrifugation at $7000 \times g$. Ni-NTA affinity chromatography was used for protein purification. Cells were resuspended in binding buffer [100 mM HEPES pH 7.5, 500 mM NaCl, 5 mM imidazole, and 5% glycerol (v/v)], then lysed with a sonicator. The insoluble cell debris was removed by centrifugation at $30,000 \times g$. The soluble cell lysate fraction was loaded on a 4 mL Ni-NTA column (QIAGEN) pre-equilibrated with binding buffer, washed with 250 mL washing buffer [100 mM HEPES pH 7.5, 500 mM NaCl, 30 mM imidazole, and 5% glycerol (v/v)], and N-terminal His6-tagged protein was eluted with elution buffer [100 mM HEPES pH 7.5, 500 mM NaCl, 250 mM imidazole and 5% glycerol (v/v)]. The His6-tagged proteins were then subjected to overnight TEV cleavage using 50 μg of TEV per mg of His6-tagged protein in binding buffer and dialyzed overnight against the binding buffer. The His6-tag and TEV were removed by re-running the protein over the Ni-NTA column. The tag-free protein was then dialyzed in crystallization buffer (50 mM HEPES pH 7.5, 500 mM NaCl) overnight, and the purity of the protein was analyzed by SDS-polyacrylamide gel electrophoresis.

**Crystallization and structure determination.** The meta-AAC0038 apoenzyme crystal was grown at room temperature using the vapor diffusion sitting drop method solution containing 20 mg/mL protein, 2.5 M ammonium sulfate, 0.1 M Bis-Tris propane pH 7, and 10 mM gentamicin. For the AG-bound structures of meta-AAC0038 and AAC(3)-IVa, we utilized the catalytically inactive mutants His168Ala and His154Ala. The meta-AAC0038[H168A]-apramycin-CoA complex was co-crystallized from solution containing 20 mg/mL protein, 20% PEG 3350, 50 mM ADA pH 7, and 10 mM apramycin. The AAC(3)-IVa apoenzyme was crystallized as selenomethionine-derivative from a solution containing 30 mg/mL protein, 0.2 M magnesium chloride, 0.1 M Tris pH 8.8, and 25% PEG3350. The AAC(3)-IVa[H154A]-apramycin complex was co-crystallized from a solution containing 0.1 M Hepes pH 7.6, 30% PEG 1 K, and 2.5 mM apramycin; the AAC(3)-IVa[H154A]-apramycin complex was co-crystallized from a solution containing 0.1 M Hepes pH 7.5, 30% PEG 1 K and 1 mM gentamicin.

Diffraction data at 100 K were collected at a home source Rigaku Micromax 007-HF/R-Axis IV system, at beamline 21-ID-G of the Life Sciences Collaborative Access Team at the Advanced Photon Source (MAR CCD detector with 300 mm plate), or beamline 19-ID of the Structural Biology Center of the Advanced Photon Source, Argonne National Laboratory. All diffraction data were processed using HKL3000[41]. For meta-AAC0038, the apoenzyme structure was solved by Molecular Replacement (MR), using the structure of YokD[34] and the CCP4 online server Balbes program. The apramycin complex structure was used solved by MR using the apoenzyme model. For AAC(3)-IVa, the apoenzyme structure was solved by MR using the structure of FrbF (PDB 3SMA)[33] and the CCP4 online server MoRDa program, and the AG bound structures were solved by MR using the apoenzyme model.

All model building and refinement were performed using Phenix.refine[42] and Coot[43]. Atomic coordinates have been deposited in the Protein Data Bank with accession codes 5HT0, 6MMZ, 6MN0, 7KES, 6MN3, 6MN4, 6MN5, 7LAO, and 7LAP. Dimerization interfaces were determined using the PDBePISA server[44]. Structural homologs were identified in the PDB using the Dali-lite server[45] or the PDBeFold server[46].

**Reporting summary.** Further information on research design is available in the Nature Research Reporting Summary linked to this article.

## Data availability
The data generated in this study are available in Supplementary Data 1 (MIC data); Protein Databank (crystal structures) under accession codes 5HT0, 6MMZ, 6MN0, 7KES, 6MN3, 6MN4, 6MN5, 7LAO, and 7LAP; or NCBI Database (i.e., FMG sampling) with accession codes indicated in Fig. 5.

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

## Acknowledgements

This work made use of the Centre for Microbial Chemical Biology core facility at McMaster University for antibiotic susceptibility testing. We thank Rosa Di Leo for cloning and Zdzislaw Wawrzak and Karolina Michalska, Argonne National Laboratory, for x-ray diffraction data collection. This project has been funded in whole or in part with Federal funds from the National Institute of Allergy and Infectious Diseases, National Institutes of Health (NIH), Department of Health and Human Services, under Contract numbers HHSN272201200026C and HHSN272201700060C. This research was supported by the Ontario Research Fund Research Excellence (ORF-RE) Grant No. RE07-048 (to G.D.W. and A.S.) and a Canadian Institutes of Health Research grant (FRN-148463) and a Canada Research Chair to G.D.W. This work is supported in part by an award to G.D. through the National Institute of Allergy and Infectious Diseases of the National Institutes of Health (U01 AI123394). A.W.D. received support from the National Research Service Award-Medical Scientist grant to Washington University (T32 GM007200), and the Institutional Program Unifying Population and Laboratory-Based Sciences Burroughs Welcome Fund grant to Washington University. The content is solely the responsibility of the authors and does not necessarily represent the official views of the funding agencies.

## Author contributions

P.J.S. solved all crystal structures, completed sequence and structural analyses, prepared figures, and wrote the manuscript. E.B. completed all antibiotic susceptibility testing, prepared figures, and wrote the manuscript. Z.X. performed protein purification and characterization. T.S. performed protein purification and crystallization. E.E. performed protein purification and crystallization. S.C. performed cloning. L.D.T. performed FMG, sequence analysis, and prepared figures. A.W.D. and S.P. performed FMG. G.D., G.D.W., and A.S. oversaw the work and wrote the manuscript.

## Competing interests

The authors declare no competing interests.
