## [Peer Review File · Communications Biology]

Reviewers' comments:

Reviewer #1 (Remarks to the Author):

Summary:

The manuscript by Stogios and co-workers presents comprehensive analyses of aminoglycoside resistance enzymes belonging to the Antibiotic_NAT family. Enzymes included in this analysis are both from clinical isolates and environmental sources. The scope of the analyses spans phylogenetic classification, antibiotic substrate spectrum studies and crystallographic structure determination. As a whole, the manuscript adds to our knowledge of what we know about enzymes belonging to the Antibiotic_NAT family.

Critique:

The title of the manuscript "structural and molecular rationale for the diversification of resistance mediated by the Antibiotic_NAT family" is intriguing and suggest that some (significant) new insights into the molecular basis of antibiotic resistance has been uncovered. Unfortunately, if that is the expectation, this manuscript does not deliver.

First, it is instructive to define what the Antibiotic_NAT family actually is. Despite what the authors state (and this has unfortunately been repeated in previous articles by these authors), the Antibiotic_NAT family is not distinct from the GNAT superfamily in either sequence or structure. A more accurate description is that the Antibiotic_NAT "family" is one of many clades in the GNAT superfamily, analogous to chihuahuas belonging to the superfamily that encompasses dogs, wolves, and coyotes. It is also instructive to state that while the name of the Antibiotic_NAT family implies a diversity of enzymes that confer resistance to different antibiotics, the reality is that for those enzymes for which a substrate is known, the substrates are only aminoglycoside antibiotics. Moreover, unlike other aminoglycoside modifying enzyme families, the Antibiotic_NAT members are essentially only able to modify one specific chemical group on this class of antibiotics (i.e. the N3 position). Thus, in summary, the Antibiotic_NAT family is a rather narrowly defined collection of enzymes.

The authors make the point several times in their manuscript that, in contrast to other aminoglycoside modifying enzymes, the enzymes of the Antibiotic_NAT family have been relatively understudied. I would disagree with this statement in so far that, while there might not be numerous publications, the work that has been reported (most notably by the group of Engin Serpersu) has been extremely thorough and has provided a wealth of information.

The new information added to our understanding of the narrowly defined collection of enzymes, as presented in this manuscript is the following: Phylogenetic analysis of an increased number of sequences now allows for a further sub classification. This is of course to be expected (as with more samples of chihuahuas we can group them into short hair and long hair breeds). The additional substrate spectrum analyses and structure determination, reveals that differences in substrate can be correlated to minor structural differences in the region that binds aminoglycosides. Again, this is of course not an unexpected result.

In conclusion, the manuscript presents a lot of data. Unfortunately, all this new data only incrementally contributes to new knowledge and does not provide new insights. Therefore, I cannot recommend considering this manuscript for publication in Communications Biology but suggest the authors submit this to a specialized journal.

Reviewer #2 (Remarks to the Author):

The manuscript describes the structural basis for aminoglycoside specificity within the antibiotic_Nat enzyme family. This is supported by excellent crystallographic data from a range of enzymes within the family. The manuscript is well written and easy to follow. I feel this is an important contribution and will influence the thinking in the field. I have only very minor suggestions for the manuscript.

Minor comments:

1. Why were the CC1/2 values not reported for all the structures?
2. Why is asterisk placed on 6MM2 CC1/2 but not other structures?

3. Supp table 2 is out of alignment in some rows, making it difficult to interpret.
4. The sentence at line 187 does not warrant being an independent paragraph.

Reviewer #3 (Remarks to the Author):

This is a well thought article that reports a comprehensive structural and functional analysis of the aminoglycoside-resistance spectrum conferred by Antibiotic_NAT family enzymes. The research has been well planned, executed, and communicated. I have only minor comments:

1. Although the introduction is quite thorough, a better description of characteristics and differences between the two families would facilitate the understanding to many readers.

2. If I understand correctly, the plasmid system is derived from pBR322. That means the copy number is around 20-25. This is what in nature is high copy number. However, with the development of pUC like replicons that multiply by 10 or more the number of copies, it seems to be low. While it is true that it is quite common to find the reference to ColE1 replicons as low copy number, I do not think it is correct. There are plasmids that have copy number 5 or 1 and low copy number should be reserved for those. However, I think that this statement should be considered as a comment because the confusion about copy numbers of plasmids is now widespread.

3. I was intrigued by the role of Zn in one of the enzymes. The presence of Zn is inhibitory for many other aminoglycoside modifying enzymes. I hope we will soon read a follow up to this article with more about its role in the activity of AAC(3)-Iva.

Response to Referees for re-submission of “Structural and molecular rationale for the diversification of resistance mediated by the Antibiotic_NAT family” by Stogios PJ, *et al*, manuscript ID COMMSBIO-21-2603-T.

Comments by Reviewer 1:

1. “The title of the manuscript “structural and molecular rationale for the diversification of resistance mediated by the Antibiotic_NAT family” is intriguing and suggest that some (significant) new insights into the molecular basis of antibiotic resistance has been uncovered. Unfortunately, if that is the expectation, this manuscript does not deliver.”

We respectfully disagree with the reviewer that our paper does not provide new insights into the molecular basis of resistance. We have adjusted some wording in Discussion to better highlight the importance of our findings.

In particular, we expanded the corresponding paragraph to:

“In this study, we follow on our previous identification of multiple Antibiotic_NAT family members in soil-derived metagenomic libraries³⁵ through detailed structural and functional analysis. Firstly, the phylogenetic reconstruction of this family that we calculated was linked to a comprehensive study of the substrate specificity profiles of the four main clades, represented by the AAC(3)-IV, AAC(3)-VII/VIII/IX/X, AAC(3)-III, and AAC(3)-II/IV enzymes. Secondly, with the additional crystal structures described in this study and comparison to previously-available structural information, we conclusively show that this division is reflected in differences in activity against AG substrates and in structural diversification localized to the minor subdomain of the Antibiotic_NAT fold. Given that the minor subdomain is much less conserved between Antibiotic_NAT family members, the deficit in molecular information about variations in this subdomain that would allow for better understanding of the role of individual amino acids in this region for substrate specificity necessitated and inspired our structural investigation into additional representatives of this family. Thirdly, we show that environment-derived enzymes of this family, which previously have not been characterized for molecular determinants behind their activity against antibiotic substrates, possess resistance-conferring activities comparable to and sometimes exceeding those activities of their counterparts derived from clinical isolates. Fourthly, we show that numerous members of this family inactivate apramycin, an atypical AG that is increasingly being considered for clinical deployment and for which little has been known about possible resistance determinants.”

2. “First, it is instructive to define what the Antibiotic_NAT family actually is. Despite what the authors state (and this has unfortunately been repeated in previous articles by these authors), the

Antibiotic_NAT family is not distinct from the GNAT superfamily in either sequence or structure. A more accurate description is that the Antibiotic_NAT “family” is one of many clades in the GNAT superfamily, analogous to chihuahuas belonging to the superfamily that encompasses dogs, wolves, and coyotes. It is also instructive to state that while the name of the Antibiotic_NAT family implies a diversity of enzymes that confer resistance to different antibiotics, the reality is that for those enzymes for which a substrate is known, the substrates are only aminoglycoside antibiotics. Moreover, unlike other aminoglycoside modifying enzyme families, the Antibiotic_NAT members are essentially only able to modify one specific chemical group on this class of antibiotics (i.e. the N3 position). Thus, in summary, the Antibiotic_NAT family is a rather narrowly defined collection of enzymes.”

While we appreciate reviewer’s comment we would like to defend our postulate about the distinct status of Antibiotic_NAT family. Our conviction is based on comprehensive comparison of the sequence, structure, and enzymatic characteristics of the two families as described in public databases. The table below summarizes the key aspects of this analysis:

Antibiotic_NAT		GNAT
Sequence databases		
PFAM	Clan Antibiotic_NAT (CL0627); PFAM Antibiotic_NAT (PF02522)	Clan Acetyltrans (CL0257); PFAM Acetyltransf_1 (PF00583)
Interpro	IPR003679 - Aminoglycoside N(3)-acetyltransferase / IPR028345 - Aminoglycoside 3-N- acetyltransferase-like	IPR016181 - Acyl-CoA N- acyltransferase / IPR000182 - GNAT domain
Structure database		
SCOP	STRUCTURAL CLASS: Alpha and beta proteins (a/b) 3 layers: a/b/a, mixed beta-sheet of 8 strands, order 78612354, strands 3, 4 and 8 are antiparallel to the rest FOLD: TTHA0583/YokD-like SUPERFAMILY: TTHA0583/YokD-like FAMILY Aminoglycoside 3-N- acetyltransferase-like	STRUCTURAL CLASS: Alpha and beta proteins (a+b) FOLD: Acyl-CoA N- acyltransferases (Nat) SUPERFAMILY: Acyl- CoA N-acyltransferases (Nat) FAMILY N- acetyltransferase, NAT

Secondary structure topology (PDBsum)	 Calculated for PDB 6MN2 (meta-AAC0038)	 Calculated for PDB 4YFJ (AAC(3)-Ib)
Sequence length	~220 residues	~120 residues
Enzymatic mechanism	Catalytic histidine activated by Glu	Catalytic acid – usually tyrosine

To better reflect the above analysis that illustrates that the Antibiotic_NAT family is indeed distinct from the GNAT superfamily we made the following addition to the Introduction of our manuscript:

These families are distinct in sequence length (approx. 120 residues for GNAT and approx. 220 residues for Antibiotic_NAT) and are classified distinctly by sequence databases (Antibiotic_NAT in Pfam: family Antibiotic_NAT (PF02522), clan Antibiotic_NAT (CL0627) vs GNAT: Acetyltransf_1 (PF00583), Clan Acetyltrans (CL0257) and by structural databases (Antibiotic_NAT in SCOP: Class = Alpha and beta proteins (a/b), Fold = TTHA0583/YokD-like, Superfamily = TTHA0583/YokD-like, Family Aminoglycoside 3-N-acetyltransferase-like vs GNAT: Class Alpha and beta proteins (a+b), Fold: Acyl-CoA N-acyltransferases (Nat), Superfamily: Acyl-CoA N-acyltransferases (Nat), Family: N-acetyltransferase, NAT). Furthermore, the distinction between these two families is reflected in the divergence in the topology of the β -sheet core of each fold, where the Antibiotic_NAT family is centered on a 3-stranded parallel β -sheet while the GNAT family is centered on a 4-stranded antiparallel β -sheet. Finally, the two families utilize distinct enzymatic mechanisms, with Antibiotic_NAT utilizing a catalytic histidine/glutamate dyad²⁹ while GNAT utilizes a catalytic tyrosine and glutamate pair³⁰.

Please note that this change in the Introduction triggered the renumbering of references throughout the manuscript.

We also want to draw reviewer's attention to the fact that, we did not make any claims about diversity in antibiotic chemical structure relating to the activity of Antibiotic_NAT family. We also did not make any claims that the Antibiotic_NAT family acetylates any position on the substrate other than the 3'-N.

3. *"The authors make the point several times in their manuscript that, in contrast to other aminoglycoside modifying enzymes, the enzymes of the Antibiotic_NAT family have been relatively understudied. I would disagree with this statement in so far that, while there might not be numerous publications, the work that has been reported (most notably by the group of Engin Serpersu) has been extremely thorough and has provided a wealth of information."*

Our comment that the reviewer is criticizing is based on the fact that the GNAT family has been much more studied than the Antibiotic_NAT family (Google Scholar search of "GNAT" retrieves about 82,600 results, search of "Antibiotic_NAT" retrieves 166). However, given the arbitrary nature of this statement we were happy to address the reviewer's comment by making the following changes:

The corresponding sentence in the Introduction has been modified to: "In contrast to the GNAT family, ~~much~~ less is known about the biochemical, structural, and molecular features of the Antibiotic_NAT family"

We also omitted the paragraph in Discussion that previously stated: "The Antibiotic_NAT family has remained understudied compared to other AME families, including AACs from the GNAT family, APHs, and ANTs. The AAC(3)-II and -III representatives of this family are widespread in pathogens, but the molecular basis for AG recognition has only recently been investigated in detail³⁰. There remain significant gaps in our understanding of the structure and function of these AMEs, from both pathogens and environmental microbes."

4. *"The new information added to our understanding of the narrowly defined collection of enzymes, as presented in this manuscript is the following: Phylogenetic analysis of an increased number of sequences now allows for a further sub classification. This is of course to be expected (as with more samples of chihuahuas we can group them into short hair and long hair breeds). The additional substrate spectrum analyses and structure determination, reveals that differences in substrate can be correlated to minor structural differences in the region that binds aminoglycosides. Again, this is of course not an unexpected result."*

We respectfully disagree with the last statement that the structural differences we observed in the minor subdomain of the Antibiotic_NAT is not an unexpected result. From a sequence perspective, this domain is much less conserved than the major subdomain and only through structural characterization presented in our manuscript the 3D conformation of this portion of the NAT_Antibiotic proteins could have been visualised and analyzed. It has been well established in structural biology, and in antimicrobial resistance research, that even single amino acid substitutions can result in differences in substrate spectrum and activity (i.e. "Twelve Positions in a β -Lactamase That Can Expand Its Substrate Spectrum with a Single Amino Acid

Substitution” *Yi et al, PLOS One, 2012*; “A structural determinant of mycophenolic acid resistance in eukaryotic inosine 5'-monophosphate dehydrogenases”, *Freedman R et al, Protein Science, 2019*). Therefore, the structures we determined provide a basis for rational understanding of the role of individual amino acids in the minor subdomain for drug specificity.

To clarify the significance of this data, we made the following change that was already mentioned in relation to comment #1:

Following sentences were added to Discussion: “Given that the minor subdomain is much less conserved between Antibiotic_NAT family members, the deficit in molecular information about variations in this subdomain that would allow for better understanding of the role of individual amino acids in this region for substrate specificity necessitated and inspired our structural investigation into additional representatives of this family.”

Comments by Reviewer 2:

1. “*Why were the CC1/2 values not reported for all the structures?*”

This was an error. The CC1/2 values have been fully entered in the updated Supplementary Data file.

2. “*Why is asterix placed on 6MM2 CC1/2 but not other structures?*”

We intended to use the asterisk to indicate the values indicate the high resolution shells for values in brackets and for CC1/2. We corrected the placement of the asterisks and updated the footnote to the table to make this more clear.

3. “*Supp table 2 is out of alignment in some rows, making it difficult to interpret.*”

This has been corrected.

4. “*The sentence at line 187 does not warrant being an indepedent paragraph.*”

This sentence has been moved to the previous paragraph.

Comments by Reviewer 3:

1. *“Although the introduction is quite thorough, a better description of characteristics and differences between the two families would facilitate the understanding to many readers.”*

We agree and the changes added to the text to respond to Reviewer 1’s similar point better explain the differences in sequence, structure and mechanism between the two families.

2. *“If I understand correctly, the plasmid system is derived from pBR322. That means the copy number is around 20-25. This is what in nature is high copy number. However, with the development of pUC like replicons that multiply by 10 or more the number of copies, it seems to be low. While it is true that it is quite common to find the reference to ColE1 replicons as low copy number, I do not think it is correct. There are plasmids that have copy number 5 or 1 and low copy number should be reserved for those. However, I think that this statement should be considered as a comment because the confusion about copy numbers of plasmids is now widespread.”*

To remove the ambiguity regarding whether our used vector (pGDP3) is a low-copy plasmid or not, in the Results section we changed this sentence: “we tested the antimicrobial susceptibility of *E. coli* individually harboring the 21 different Antibiotic_NAT genes on a low-copy plasmid”
To:

“we tested the antimicrobial susceptibility of *E. coli* individually harboring the 21 different Antibiotic_NAT genes on the pGDP3 plasmid”.

Based on the cited reference #35, the reader will judge whether this is a low-copy plasmid.

3. *“I was intrigued by the role of Zn in one of the enzymes. The presence of Zn is inhibitory for many other aminoglycoside modifying enzymes. I hope we will soon read a follow up to this article with more about its role in the activity of AAC(3)-Iva.”*

We are also intrigued by the role of Zn in the Group 1 enzymes including AAC(3)-IVa and thank the reviewer for their enthusiasm. The Discussion did include some remarks on what has been done to study this ion and its coordinating residues, and we plan to conduct further experiments to investigate this. At this point in time, the only additional data we have is that single point mutations of the 4 Zn²⁺ coordinating residues (3 Cys and 1 His residue) in AAC(3)-IVa to alanine do not affect the solubility of the recombinant enzyme from *E. coli*. We did not produce the double mutant Cys247Ser/Cys250Ser that was discussed. Thus we have no further comments to add to the manuscript and look forward to future results in this area.